# Field-based detection of bacteria using nanopore sequencing: Method evaluation for biothreat detection in complex samples

**Andrea D. Tyler**[1], **Jane McAllister**[2], **Helen Stapleton**[3], **Penny Gauci**[2], **Kym Antonation**[1],
**David Thirkettle-Watts**[2], **Cindi R. Corbett**[1,4]*

1 National Microbiology Laboratory, Public Health Agency of Canada, Winnipeg, Manitoba, Canada,
2 Department of Defence, Defence Science and Technology Group, Fishermans Bend, Melbourne, VIC,
Australia, 3 Defence Science and Technology Laboratory, Dstl Porton Down, Salisbury, Wiltshire, United
Kingdom, 4 Department of Medical Microbiology and Infectious Diseases, University of Manitoba, Winnipeg,
Manitoba, Canada

* cindi.corbett@canada.ca

pone.0295028

Arab Emirates University, UNITED ARAB
EMIRATES

**Data Availability Statement:** Samples available as
part of NCBI BioProject: PRJNA943266.

## Abstract

From pathogen detection to genome or plasmid closure, the utility of the Oxford Nanopore
Technologies (ONT) MinION for microbiological analysis has been well documented. The
MinION's small footprint, portability, and real-time analytic capability situates it well to
address challenges in the field of unbiased pathogen detection, as a component of a secu-
rity investigation. To this end, a multicenter evaluation of the effect of alternative analytical
approaches on the outcome of MinION-based sequencing, using a set of well-characterized
samples, was explored in a field-based scenario. Three expert scientific response groups
evaluated known bacterial DNA extracts as part of an international first responder (Chemi-
cal, Biological, Radiological) training exercise. Samples were prepared independently for
analysis using the Rapid and/or Rapid PCR sequencing kits as per the best practices of
each of the participating groups. Analyses of sequence data were in turn conducted using
varied approaches including ONTs What's in my pot (WIMP) architecture and in-house
computational pipelines. Microbial community composition and the ability of each approach
to detect pathogens was compared. Each group demonstrated the ability to detect all spe-
cies present in samples, although several organisms were detected at levels much lower
than expected with some organisms even falling below 1% abundance. Several 'contami-
nant' near neighbor species were also detected, at low abundance. Regardless of the
sequencing approach chosen, the observed composition of the bacterial communities
diverged from the input composition in each of the analyses, although sequencing con-
ducted using the rapid kit produced the least distortion when compared to PCR-based
library preparation methods. One of the participating groups generated drastically lower
sequencing output than the other groups, likely attributed to the limited computer hard drive
capacity, and occasional disruption of the internet connection. These results provide further
consideration for conducting unbiased pathogen identification within a field setting using
MinION sequencing. However, the benefits of this approach in providing rapid results and
unbiased detection must be considered along with the complexity of sample preparation

**Funding:** This work was supported by the Canadian Safety and Security Program project, CSSP-2018-TI-2372. CR Corbett, KS Antonation were recipients of the CSSP funding. The funders had no role in study design, data collection and analysis, decision to publish, or preparation of the manuscript. Note the opinions expressed within do not represent the opinions of the Public Health Agency of Canada or the Government of Canada.

**Competing interests:** The authors have declared that no competing interests exist.

and data analytics, when compared to more traditional methods. When utilized by trained scientific experts, with appropriate computational resources, the MinION sequencing device is a useful tool for field-based pathogen detection in mixed samples.

## Introduction

In the rapidly evolving field of bioforensics and biothreat response, application of novel methods to analyze various samples in non-laboratory settings is critical. The capability to detect a wide variety of pathogens without *a priori* knowledge of the potential threat agent in a non-laboratory setting is the ideal. As such, Oxford Nanopore Technologies (ONT) MinION device, which can rapidly produce long read sequence data for agnostic biothreat detection and characterization, is ideally situated: it is small and highly portable, generates long read sequence data, has the ability to analyze reads almost immediately following the start of a sequencing run, and can be used for a variety of sample types, including mixed environmental samples [1–3]. Previous work has demonstrated this technologies' utility in the context of field-based detection of Ebola, Zika, and tuberculosis (TB), and it has been previously shown to have use in remote settings simulating biothreat response [4–6]. Despite this body of work and several sources describing the strengths and limitations of this method [7–9], none to date have documented the capability of the MinION device on a set of well characterized samples in the field, as part of a blinded, multicentre assessment. To this end, teams from Australia, Canada and the United Kingdom evaluated the capability of the MinION sequencing platform for identification of bacterial agents of interest in a set of mixed DNA samples as part of an international first responder exercise.

## Materials and methods

Prior to the exercise, sample DNA was prepared by Group 1 as follows: Bacterial strains were incubated at 37˚C overnight in LB broth, and DNA was extracted using the Blood and Cell Culture DNA Midi kit (Qiagen), with final elution of DNA in Tris buffer. Strains included in the analysis were: *Bacillus anthracis* Sterne 24F2 (3ng/uL), *Bacillus thuringensis* kurstaki (36 ng/uL), *Burkholderia cepacia* ATCC25416 (484 ng/uL), *Escherichia coli* PA03M55684 (60 ng/uL), *Enterococcus faecalis* 159905163 (14 ng/uL), *Staphylococcus aureus* ATCC9144 (14 ng/uL), *Yersinia pseudotuberculosis* ATCC 13979 (324 ng/uL) (Table 1). All DNA was of high quality (Nanodrop 2000 –Thermo Scientific & Bioanalyzer—Agilent) and concentrations were determined using Qubit (Thermofisher). Pure DNA was then used to create pools of mixed bacterial communities for sequencing as described in Table 1. Group 2 also sequenced each of the pure gDNA samples from the individual isolates, with this data used to explore the rate at

**Table 1. Mixed unknown samples used in the comparative community analysis.** (Genomic equivalents in millions).

| Pool A | Pool B | Pool C | Pool D |
|---|---|---|---|
| *B. cepacia* (224M) | *S. aureus* (46M) | *Y. pseudotuberculosis* (30M) | *Y. pseudotuberculosis* (61M) |
| *Y. pseudotuberculosis* (150M) | *E.coli* (36M) | *B. cepacia* (26M) | *E. coli* (12M) |
| *B. thuringiensis* (67M) | *B. cepacia* (26M) | *E.coli* (12M) | *B. thuringiensis* (2.8M) |
| *E. faecalis* (32M) | *B. anthracis* (16M) | *B. thuringiensis* (5.6M) | *B. anthracis* (2.5M) |
| *V. parahaemolyticus* (19M) | | *S. aureus* (4.6M) | *B. cepacia* (1.3M) |
| | | *E. faecalis* (3.8M) | |
| | | *V. parahaemolyticus* (1.8M) | |
| | | *B. anthracis* (0.8M) | |

which data was generated over time. Groups 2 and 3 were blinded to the composition of samples.

Samples were processed as described in S1 Fig. In brief, each group applied their best practice library preparation protocols and sequence analysis methods to the prepared samples. Sequencing library preparation kits used by each of the groups included the Rapid barcoding (SQK-RBK004; RSE_9046_v1_revW_14Aug2019), and Rapid PCR barcoding kits (SQK-RPB004; RPB_9059_V1_REVA_08MAR2018)(Oxford Nanopore Technologies, Oxford UK). Sequencing was performed using ONT MinION flowcells (R9.4.1) and Mk1B sequencing devices. Flowcells and reagents used by Groups 2 and 3 were transported to the location of the field exercise via commercial aircraft. As such, while an attempt was made to maintain equipment and reagents at appropriate temperatures, they may have been exposed to prolonged thawing/warming and changes in pressure.

Flowcells were assessed for and passed minimum QC metrics at the time of sequencing. Library preparation and sequencing were carried out as per protocols provided by ONT. As this exercise was designed to simulate sequencing in the field, minimal protocols were conducted: no additional DNA quantification was performed by any of the groups prior to library preparation, rather each sample was added using the maximum volume as indicated by the ONT protocol, and AMPure bead cleanup was omitted following barcoding.

Both Groups 1 and 3 made use of Windows-based software products produced and marketed by ONT on systems described in Table 2. Group 2 performed their analysis on an Intel Server, with customized software for data analysis, which included use of albacore (v 2.3.1) (ONT, Oxford UK) and Kraken (1.0) with a reference database consisting of bacterial, viral, archaeal, fungal and protist species, obtained from NCBI's RefSeq collection (2018-April-20) (Table 2) [10]. An abundance threshold for detection and reporting of 1% is used by default by the What's in my pot (WIMP) analysis pipeline, as such although not part of standard protocol and in order to maintain consistency between methods, Group 2 also applied an abundance threshold cutoff of 1% to the taxonomic analysis.

## Results

Each of the groups were able to successfully sequence all of the mixed microbial community samples included as part of this analysis. Group 3 had challenges with getting runs started, due

**Table 2. Comparison of computational resources used by each of the participating partners for data capture and analysis.**

|  | Recommended by ONT | Group 1 | Group 2 | Group 3 | MinIT (for comparison) |
|---|---|---|---|---|---|
| Platform | Not specified | Intel NUC | LeNovo ThinkServer | Dell Latitude 5580 | MinIT |
| Operating system | Windows/ Ubuntu/Sierra | Windows | Ubuntu | Windows | Linux |
| Memory | 16GB RAM | 32GB RAM—max | 504GB RAM | 16G RAM | 8GB RAM/GPU accelerators |
| CPU | i7 or Xeon with 4 + cores | i7, 3.5GHz—max | Intel Xeon E5- 2.20 GHz | Intel R Core i7-7820HQ | Unclear |
| Storage | 1TB SSD | 2TB max | 3TB (mirrored) | 1TB | 512GB |
| USB3 Ports | Yes | Yes | Yes | Yes | Yes |
| Internet connection | Required | Required for WIMP only | Not Required (ping free) | Required for MinKnow and WIMP (need to ping) | Not Required |
| Basecalling Software | MinKNOW/ Albacore | MinKNOW (2.2) | Albacore (2.3.0) | MinKNOW (2.2) | MinKNOW |
| Bacterial identification software | WIMP | WIMP (3.2.0) | Kraken (1.0)–custom database; signature sequence analysis | WIMP (3.2.0) | Optional |

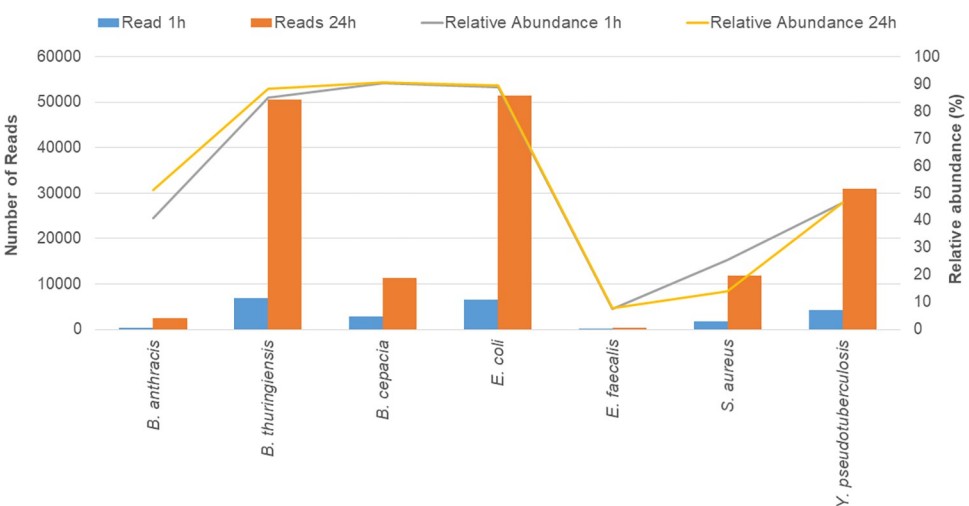

**Fig 1. Comparison of identification of species from the pure DNA extracts 1 hour and 24 hours following the start of sequencing, performed by Group 2.**

to unstable internet connectivity, which is a requirement of the off-the-shelf MinKNOW software. This group also ran into challenges with data storage and slow basecalling, leading to a reduced amount of data being generated. It is unclear whether these challenges arose due to their computational setup, in which they were just meeting the minimum requirements suggested by ONT, or due to unstable internet connectivity and a lack of access to ping-free software.

Analysis of results obtained 1 and 24 hours following the start of sequencing of the single organism samples (Fig 1), demonstrated that DNA sequence could be assigned to species one hour following the beginning of sequencing. Importantly, the abundance of accurately classified organisms detected in the sample did not substantially change between time points. As sequences are generated immediately upon sample addition to the sequencer, the limitation in time to detection is in the transfer of data to the analysis cloud for samples analyzed using the Epi2Me WIMP workflow (Oxford Nanopore Technologies), and in the time taken to both basecall and taxonomically assign reads, rather than the sequencing itself. This suggests that advancements in computational approaches will reduce the time to detection of organisms of interest.

Results of the sequence analysis of the mixed bacterial pools analyzed by each of the groups after 24 hours of sequencing are included in Fig 2. Group 1 generated a large number of sequences throughout the run, and despite use of the Rapid sequencing protocol, had a similar number of reads detected to that generated by Group 2 when using the Rapid PCR protocol. It is unclear as to the reason for this observation as typically PCR-reliant protocols result in increased sequencing yield, it is possible that the transportation of reagents to the location of the exercise by each of the groups may have led to inconsistent efficiency (Group 2 required international travel, and Group 1 domestic travel). The number of reads generated and analyzed by Group 3 were much lower than either of the other groups, which was attributed to the computational challenges faced by this group, and potentially the transport of equipment and reagents to the exercise location.

All organisms from each of the mixed samples were successfully identified by both groups 1 and 2, although several fell below the 1% abundance threshold used for the analysis. Interestingly, each group observed that community composition varied from the input, with *B.*

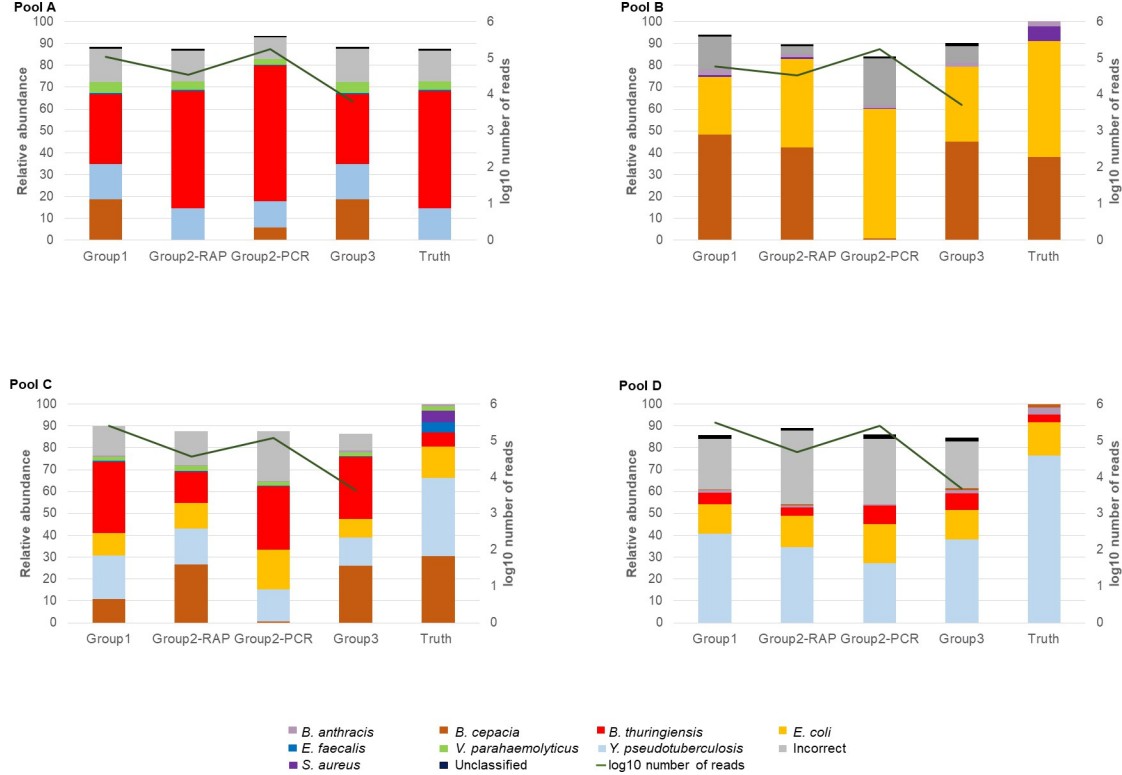

**Fig 2. Relative abundance of different species detected in each of the pools by each of the groups following 24 hours of sequencing.** Group 2 performed the analysis using both the Rapid (SQK-RBK004—RAP) and Rapid PCR (SQK-RPB004—PCR) barcoding sequencing kits, on separate flowcells. The log10 number of reads detected in each of the samples is described along the right axis. The Rapid PCR sequencing conducted by Group 2 produced so many reads, it was stopped after 8 hours of sequencing.

*thuringiensis* commonly over represented in samples, and *E. faecalis*, *B. cepacia*, *S. aureus* and *V. parahaemolyticus* commonly under-represented. Notably, *B. cepacia* was detected in a manner more in keeping with the input sample when the Rapid sequencing kit without PCR amplification was used. Incorrect species which were present at >1% relative abundance included both near neighbour species likely arising from both misclassification (*Y. similis* was observed in Pool C by each of the groups, and was observed when Group 2 used both the Rapid and Rapid PCR kits, likely due to its homology with *Y. pseudotuberculosis*) and cross-contamination during the preparation of the sample or sequencing library (*B. thuringiensis* was detected in all pooled samples, including those for which none of this organism was added, and at levels disproportionate to the input amount). Reads which were correctly assigned to taxonomic levels above species, were not included in reporting.

## Discussion

The results of this analysis have demonstrated that nanopore-based sequencing has utility in the rapid detection and analysis of both pure and mixed samples in a field deployment scenario, when supported by scientific expertise and appropriate on-site analytical capabilities. This includes, ensuring strong internet connectivity and having available sufficient computing power for identification of sequence reads. While each group approached the analysis of samples and data differently, similar results were obtained, suggesting that consistent results,

especially in the realm of pathogen detection, are possible across methodologies. Importantly, analytical methods used to evaluate microbial composition did not perform substantially differently in terms of output between groups, with user-friendly cloud-based detection methods (WIMP) performing equally to customized pipelines.

An important caveat to this was the experience reported by Group 3, who, while using a computational setup which met the minimum requirements set out by ONT, had trouble both getting runs to start, and produced less sequencing data than did the other groups. Importantly, despite the close physical proximity to other groups, there may have been inconsistent access to wireless internet which hindered this group's ability to perform sequencing. Thus, *a priori* knowledge of the availability of stable internet remains an important consideration for investigators conducting field-based analyses, both at the sequencing stage, and when performing cloud-based analysis. Other tools (MinIT, Mk1C) are capable of sequencing independent of a wireless signal, and are worth investigating for use in scenarios in which internet stability cannot be guaranteed. Alternatively, use of ping free software and locally available computational hardware exceeding the minimum specifications set out by ONT for intensive analyses, also improve reliability and performance. Use of local analysis pipelines also provides users with analytical flexibility, in the event that available protocols do not meet user needs.

Importantly, despite similarities in results between the groups, bacterial quantification in mixed samples by each group was disparate from true sample composition, with no methodology exactly replicating the proportion of reads that were in the initial sample. Such findings are consistent with microbiome sequencing performed by other methods [11], and are speculated to primarily arise from extraction, amplification and sequencing bias produced by the experimental process [12,13]. Such biases are important to consider in designing appropriate universal detection approaches, as pathogenic organisms may be missed when inappropriate analyses are used. This demonstrates that while metagenomic sequencing approaches may be universal in design, in actuality there are limitations which must be considered prior to experimental analysis when hoping to utilize a one-size-fits-all approach. All kits are not created equal and an understanding of the limitations and bias of each is vital to inform operational utility. This is particularly relevant in the event that such methods might be used by field responders who may not have expertise or understanding in the fundamental use of sequencing tools for detection and identification purposes. In addition, including techniques such as genomic or targeted amplification or multiple extractions to target different classes of bacteria may also be beneficial to increase sensitivity [14–17], but have the disadvantage of increasing turn-around-time and being less easily deployable in the field. Perhaps unsurprisingly, sequencing methods which involve less experimental manipulation (rapid sequencing kit–SQK-RBK004) appear to generate results which more closely approximate the input community structure. Thus, in cases in which abundant DNA is available, availability of a sequencing method which does not require additional manipulation or PCR is useful for detecting species which are less amenable to amplification.

While this study was conducted on a small, well-defined dataset, it has highlighted the utility of MinION sequencing in the field, for presumptive identification of bacterial species in mixed samples. Further, applying this as a rapid triage tool to understand microbial composition within a sample can direct the use of targeted, confirmatory testing while in a field scenario. While our results suggest that experimental outcomes are robust to different methodologies, it is important to note that in each case, operation of equipment, and interpretation of data by experienced laboratory personnel remains imperative. Importantly, this work adds to the body of literature supporting the use of ONT's MinION sequencing device for field-based projects, particularly in the context of rapid pathogen identification.

## Conclusion

The data presented in this manuscript illustrates the utility of nanopore-based sequencing for in-field detection of bacterial pathogens in mixed samples. The results of this work emphasize that despite all species present in the input being detected in the output sequencing data, alterations in the observed structure of the microbial community were ubiquitous and could confound interpretation by untrained users. Not surprisingly, sequencing methods which did not rely on PCR amplification for DNA analysis showed less sequencing bias than did methods which employed amplification approaches. Such observations must be balanced with the need to maximize sequencing output.

## Supporting information

**S1 Fig. Workflow depicting analytic strategy used for this analysis.**
(PDF)

## Author Contributions

**Formal analysis:** Andrea D. Tyler, Jane McAllister, Helen Stapleton.

**Funding acquisition:** Cindi R. Corbett.

**Methodology:** Jane McAllister, Helen Stapleton, Penny Gauci, Kym Antonation.

**Project administration:** Kym Antonation.

**Resources:** Jane McAllister, Helen Stapleton, David Thirkettle-Watts, Cindi R. Corbett.

**Software:** Andrea D. Tyler.

**Supervision:** Cindi R. Corbett.

**Writing – original draft:** Andrea D. Tyler.

**Writing – review & editing:** Jane McAllister, Helen Stapleton, Penny Gauci, Kym Antonation, Cindi R. Corbett.

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
