## [Decision Letter · Decision Letter 0]

25 Jun 2023

PONE-D-23-14145Field-based detection of bacteria using nanopore sequencing: method evaluation for biothreat detection in complex samples.PLOS ONE

Dear Dr. Tyler,

Thank you for submitting your manuscript to PLOS ONE. After careful consideration, we feel that it has merit but does not fully meet PLOS ONE’s publication criteria as it currently stands. Therefore, we invite you to submit a revised version of the manuscript that addresses the points raised during the review process.

We look forward to receiving your revised manuscript.

Kind regards,

Farah Al-Marzooq, MD, PhD

Academic Editor

PLOS ONE

“This work was supported by the Canadian Safety and Security Program project, CSSP-2018-TI-2372”

Additional Editor Comments:

Please revise the manuscript as advised by the reviewers

Reviewers' comments:

Reviewer's Responses to Questions

**Comments to the Author**

1. Is the manuscript technically sound, and do the data support the conclusions?

Reviewer #1: Yes

Reviewer #2: Yes

2. Has the statistical analysis been performed appropriately and rigorously? 

Reviewer #1: Yes

Reviewer #2: Yes

3. Have the authors made all data underlying the findings in their manuscript fully available?

Reviewer #1: Yes

Reviewer #2: Yes

4. Is the manuscript presented in an intelligible fashion and written in standard English?

Reviewer #1: Yes

Reviewer #2: Yes

5. Review Comments to the Author

Reviewer #1: -It is confusing without the line numbers and page numbers to follow and write the comments.

-What was the main conclusion? It needs to be clearly written in the abstract

-Abstract: Line 12: WIMP: acronyms should be written in full upon the first appearance.

-Abstract: Line 3: proofread the grammar: “MinIONs small footprint”.

-Abstract: Line 16: remove the comma.

-In the impact statement: this sentence is not understood and needs to be re-written: “showed less in the way of sequencing bias”. And the impact statement needs to be clear about the mai conclusions of the study.

-The resolution of Figure 1 needs to be better to be read easily. Can the scientific names of the strains on the x axis be italicized? Also figure 2 labels need to be larger and with better resolution.

-Grammatical proofreading is needed throughout the text.

Reviewer #2: The manuscript describes a procedure that will be the alternative or new technique for the pathogen or disease detection in samples. More study should be done for the validation of the methods but this paper it's a good start. I suggest adding a figure describing/outlining the workflow of the samples each group analysed.

6. PLOS authors have the option to publish the peer review history of their article (what does this mean?). If published, this will include your full peer review and any attached files.

Reviewer #1: No

Reviewer #2: No

---

## [Author Response · Author response to Decision Letter 0]

2 Oct 2023

Reviewer #1: -It is confusing without the line numbers and page numbers to follow and write the comments.

-What was the main conclusion? It needs to be clearly written in the abstract

-Abstract: Line 12: WIMP: acronyms should be written in full upon the first appearance.

-Abstract: Line 3: proofread the grammar: “MinIONs small footprint”.

-Abstract: Line 16: remove the comma.

-In the impact statement: this sentence is not understood and needs to be re-written: “showed less in the way of sequencing bias”. And the impact statement needs to be clear about the mai conclusions of the study.

Thank you for these valuable notes. We have addressed those specifically listed here and have revised the rest of the manuscript for grammatical edits as per this recommendation. 

-The resolution of Figure 1 needs to be better to be read easily. Can the scientific names of the strains on the x axis be italicized? Also figure 2 labels need to be larger and with better resolution.

Thanks to the reviewer for pointing this out. We have revised the images with greater resolution, and are anxious to hear whether we have addressed this concern sufficiently. 

-Grammatical proofreading is needed throughout the text.

We have addressed the specific grammatical issues listed above and have revised the rest of the manuscript for grammatical edits as per this recommendation. 

Reviewer #2: The manuscript describes a procedure that will be the alternative or new technique for the pathogen or disease detection in samples. More study should be done for the validation of the methods but this paper it's a good start. I suggest adding a figure describing/outlining the workflow of the samples each group analysed.

We have included a summary flowchart outlining our sample handling processes as part of Supplementary Figure 1.

---

## [Editor Report · Decision Letter 1]

14 Nov 2023

Field-based detection of bacteria using nanopore sequencing: method evaluation for biothreat detection in complex samples.

PONE-D-23-14145R1

Dear Dr. Tyler,

We’re pleased to inform you that your manuscript has been judged scientifically suitable for publication and will be formally accepted for publication once it meets all outstanding technical requirements.

Kind regards,

Farah Al-Marzooq, MD, PhD

Academic Editor

PLOS ONE
---

## [Editor Report · Acceptance letter]

16 Nov 2023

PONE-D-23-14145R1 

Field-based detection of bacteria using nanopore sequencing: method evaluation for biothreat detection in complex samples. 

Dear Dr. Tyler:

I'm pleased to inform you that your manuscript has been deemed suitable for publication in PLOS ONE. Congratulations! Your manuscript is now with our production department. 

Kind regards, 

on behalf of

Dr. Farah Al-Marzooq 

Academic Editor

PLOS ONE